# Well-Being, Mental Health, and Study Characteristics of Medical Students before and during the Pandemic

**DOI:** 10.3390/bs14010007

**Published:** 2023-12-21

**Authors:** Alexandra Huber, Luna Rabl, Thomas Höge-Raisig, Stefan Höfer

**Affiliations:** 1Department of Psychiatry, Psychotherapy, Psychosomatics and Medical Psychology, Medical University of Innsbruck, 6020 Innsbruck, Austria; 2Department of Psychology, University of Innsbruck, 6020 Innsbruck, Austria; luna.rabl@uibk.ac.at (L.R.); thomas.hoege@uibk.ac.at (T.H.-R.)

**Keywords:** well-being, mental health, study characteristics, medical students, pandemic, response shift effects

## Abstract

Medical students report high demands, stressors, pressure to perform, and a lack of resources, and are therefore at a higher risk for mental strain and burnout. Due to the COVID-19 pandemic, study conditions have changed, and new challenges have emerged. Thus, this study aimed to examine medical students’ well-being, mental health, and study characteristics before and during the pandemic. Data from 988 Austrian medical students were included into the cross-sectional comparisons, and 63 students were included into the longitudinal analyses (variance analyses/*t*-tests or appropriate non-parametric tests). Well-being before and during the pandemic did not differ significantly but the peri-pandemic cohort reported higher study satisfaction, more social support from lecturers, and less emotional exhaustion, cognitive demands, and stressors (information problems, organizational stressors, work overload). Longitudinally, work overload was also perceived to be higher before the pandemic; however, study satisfaction was lower. During the pandemic, approximately every seventh student exceeded the cut-off value for generalized anxiety disorder, and approximately every tenth student exceeded the cut-off value for major depression. These unexpected peri-pandemic results concerning constant high well-being, study satisfaction, and the perception of conditions may be based on response shift effects that require further exploration. The scores exceeding the reasonable cut-point for identifying probable cases of generalized anxiety disorder and depression may reflect medical students’ needs, calling for an in-depth analysis if further health promotion is necessary.

## 1. Introduction

The appearance of the coronavirus disease in 2019 (COVID-19) was described as “a public health emergency of international concern” by the World Health Organization (WHO) on 30 January 2020 and then declared as a pandemic on the 11 March 2020 [1]. In Austria, the first COVID-19 infections were registered on the 25 February 2020 [2]. The threat and fear of a rapid spread of the virus and the perceived inability to treat the infection lead many governments worldwide to implement restrictive measures to avoid further spreading, such as “strict lockdowns”, with the first one in Austria starting on the 16th of March 2020 [3]. Cross-country investigations on mental health of the general public since the beginning of the pandemic (e.g., in Spain, Italy, Iran, the USA, Turkey, Nepal, and Denmark) consistently revealed lower psychological well-being and higher rates of negative mental health outcomes, such as loneliness, anxiety, and depressive symptoms [4,5]. Even before the pandemic, medical students often reported high demands, stressors, pressure to perform, and a lack of resources, thus experiencing reduced levels of well-being and being at a higher risk of burnout and mental strain including stress, anxious/depressive symptoms, alcohol and substance abuse, as well as suicidal ideation [6,7,8,9,10]. Before studying medicine at a public Medical University in Austria, individuals have to pass the comprehensive admission test “MedAT” which takes place once a year. Both first-time admission and continuation of medical studies require German language skills at a C1 level [11]. The first study year, being thoroughly structured and following a tight schedule prescribed by the institution, starts with a heavy workload from theoretical and quite comprehensive courses (anatomy, chemistry, histology, physics, and physiology). In the second and third years, this knowledge will be deepened and complemented by problem-oriented learning as well as more courses focusing on practical and communication skills. The last three years concentrate mostly on clinical courses and bedside teaching as the main part of the semester hours [12]. The workload throughout the medical curriculum is high but sometimes verbalized as more stressful by the students in the beginning when learning all the new content compared to applying the acquired knowledge later on. Due to the pandemic, study conditions quickly had to be adapted to the unprecedented situation, and new challenges emerged [3]. Studies have suggested that medical students around the world have experienced significant stress as a result of the pandemic or measures implemented, including concerns about their training, increased workloads, and exposure to the virus [13,14,15,16]. Against this background, the aim of this study was to document and explore the impact of the management of the COVID-19 pandemic on Austrian medical students in terms of their well-being and mental health. 

### 1.1. COVID-19 in Austria

At the beginning of the pandemic, the measures in Austria included primarily non-pharmaceutical interventions such as physical distancing, wearing face masks in various specifications depending on the location (e.g., FFP2 masks in hospitals), and repeated lockdowns (curfews) or shutdowns (e.g., universities) to varying degrees. In the Austrian higher education sector, face-to-face classes were mostly prohibited right from the start [3]. Further into the pandemic, the promotion of vaccination was added and supported by different measures of social isolation of non-vaccinated people up to a governmental vaccination mandate, which in the end, was never implemented in Austria [17]. In particular, the exclusion of taking part in regular socializing without a mandated vaccine was one of the strictest and most controversial measures in Austria. All these measures also affected medical students in the summer term of 2020, when many lectures were either canceled or swiftly transferred to online lectures/seminars or, wherever possible, into seminars with a dramatically reduced number of students [3]. Traditional formats such as “bedside teaching” had to be canceled as well due to the restrictions in hospitals that only allowed trained health professionals to enter. The uncertainty about and measures dealing with the pandemic have caused a surge in negative psychological outcomes, which has been observed globally [18]. In particular, healthcare workers, being at the forefront of the pandemic, have faced increased workloads and potential exposure to the virus, which have led to higher levels of burnout and mental health problems [19]. Thus, social support was a crucial factor enabling people to buffer the negative effects of the pandemic—having access to emotional support, social connections, and practical assistance was helpful in coping with the stressors caused by the pandemic [20]. In contrast, social isolation and loneliness have been found to increase the risk of anxiety and depression [21]. With respect to Austrian university students, empirical data, especially from the beginning of the pandemic, indicated a significant decrease in well-being and social integration during the lockdown periods [22]. Although an adaptation of Austrian students to distance learning modalities during the course of the pandemic was observed by Bork-Hüffer and colleagues [3], a majority desired more face-to-face interactions for social benefits. The authors also reported an increase in headaches, dorsal pain, and tired eyes due to sitting in front of a computer for long periods of time.

### 1.2. Mental Health

According to the definition of the WHO, mental health is “... a state of mental well-being that enables people to cope with the stresses of life, realize their abilities, learn well and work well, and contribute to their community. It is an integral component of health and well-being that underpins our individual and collective abilities to make decisions, build relationships and shape the world we live in. Mental health is a basic human right. And it is crucial to personal, community and socio-economic development” [23]. Determinants of well-being and mental health include age, sex, race, education, income, employment, social relationships, marriage, friendships, leisure, volunteering, social roles, religion, genetic predisposition, personality traits, self-esteem, personal control, optimism, and goals [24]. In the WHO definition, mental health is extended to a broad, complex continuum with varying degrees of difficulty and distress, potentially leading to very different social and clinical outcomes [23]. However, assessments concerning mental health in medical settings still seem to focus more on indicators of ill-being such as anxiety or depressive symptoms rather than well-being. The global prevalence rates of anxiety and depression in medical students before the pandemic were approximately 34% and 28%, respectively, with the rates in European countries being at approximately 28% [9] and 20% [25]. These global rates increased during the pandemic up to 38% for anxiety (Europe: 42%) and 41% for depression (Europe: 43%), with the highest prevalence for psychological distress reaching 58% (Europe: 49%) [26]. Even before the pandemic, the available estimates for these rates were between two and five times higher than those reported in the age-matched general population [10]. The Austrian project “Stress in Students”, using data from December 2020 to June 2021, found high prevalence rates for various mental health conditions in Austrian students in terms of depression (35%), anxiety (29%), alcohol abuse (21%), somatoform (21%), and eating disorders (9%), as well as high levels of stress in general [27]. Moreover, 44% of the students in this project reported being “poor sleepers” with another 21% reporting scores indicative of chronic sleep disturbance. Moreover, medical students who lived with their families during the pandemic assessed their stress level as two and a half times more severe compared to students who lived alone, whereas the family status (single vs. partnered) did not have any influence [28]. In contrast, the prevalence rates of mental health disorders in terms of well-being are less well documented before and during the pandemic. Conclusions regarding well-being are mostly drawn from the absence of symptoms/diseases rather than through the use of appropriate well-being measures. Moreover, the measurement of well-being is impaired by the use of many heterogenetic instruments.

### 1.3. Thriving

Thriving comprises a broad range of well-being constructs covering psychological (PWB) and subjective well-being (SWB), thereby representing a holistic view of positive functioning [29]. Individuals’ positive potentials can contribute to this optimal functioning in terms of physical, mental, and social health [30]. In particular, PWB emerges from realizing all the potentials an individual pursues to live life in a self-fulfilling way [31]. Thriving takes seven theoretical core dimensions into account: (1) SWB, (2) enriching and supportive relationships, (3) engagement and interest in daily activities, (4) meaning and purpose in life, (5) mastery and accomplishment, (6) feelings of autonomy and control, and (7) optimism. SWB is associated with the hedonistic tradition, where well-being emerges when physical, intellectual, or social needs are satisfied and accompanied by positive emotions including happiness, enjoyment, and pleasure [31,32]. Thus, satisfaction with life in different domains and the presence of positive and the absence of negative affect form the construct of SWB [33], which is also relevant for thriving. Previous longitudinal studies have shown that the level of medical students’ well-being (means in the upper third of the range) remained stable before the pandemic [34,35]. However, the majority of medical students felt a deterioration of their well-being during the pandemic period [36], despite being already burdened prior to the pandemic by diminished physical, mental, and psychosocial well-being [37]. A German study determined that 72.2% of the students were “strongly impaired” in terms of their well-being [38].

### 1.4. Burnout

The construct of burnout is considered a three-dimensional psychological syndrome of emotional exhaustion (reduced emotional/internal resources, feelings of having nothing more to give to the job), depersonalization/cynicism (trying to distance oneself from the job, increasing cynicism about the value of work, starting to actively ignore positive aspects of the job), and reduced personal accomplishment/inefficacy (feelings of less effectiveness in the job, performance decreases) [39,40]. According to the Job Demands–Resources (JD-R) model [41], the symptoms of burnout can occur when people experience an imbalance between impeding job demands (e.g., work overload, time pressures) and facilitating resources (e.g., supportive colleagues, autonomy, feedback). This model, emanating from work and organizational psychology, is also applicable in the context of health professional studies [42]. Unfavorable circumstances in the learning environment of medical education (highly demanding study conditions while condition-related resources often appear to be lacking) were held responsible for the development of study burnout by some researchers [43]. Ongoing academic stressors exist throughout the whole medical student period, with an increasing scholastic workload, competition to achieve high scores in order to obtain the career opportunities, multiple high-stake examinations, and expectations of teachers and parents. During clinical internships, unfamiliar environments, poor teaching, limited supervision and mentoring, and poor role models all affect functioning and engagement [44]. A systematic review estimated that at least one-half of all medical students may have been affected by burnout during their education, even before the pandemic [45]. Recently published meta-analyses (partly including peri-pandemic data) reported on the prevalence of burnout rates, which depended on age, sex, and country as well as used questionnaires, with an overall range from 7% to 75.2% [46], including 38.1% to 40.8% for “emotional exhaustion”, approximately 35.1% for “depersonalization”, and 27.4% to 37.2% for “reduced personal accomplishment” [47,48].

### 1.5. Study Characteristics: Demands, Resources, Stressors

In work contexts, the predictive impact of job demands, stressors, and resources on mental health and well-being is well established [49,50]. Based on Action Regulation Theory (ART) [51,52], a structural equivalence between the conceptualization of psychologically relevant work and study characteristics regarding their effects on individuals’ mental health and well-being can be assumed [53]. According to ART, goal-oriented actions that are psychologically regulated by individuals form the core of “work”, whether these actions are paid or unpaid. Similar to employees in waged labor, students have to accomplish tasks with specific characteristics and under specific conditions which can either facilitate or impede their regulation of psychological actions, with subsequent positive or negative effects on aspects such as motivation, health and well-being, and personal growth. One prominent model of work characteristics affecting health and well-being is the above-mentioned JD-R model [41]. Another model integrates aspects of the JD-R model, ART, and the Challenge–Hindrance Stressors framework [54], distinguishing three classes of work characteristics with respect to their hypothesized impacts on motivation, growth, well-being, and health: learning demands, work stressors, and work resources. Applying these conceptual perspectives to the “work” of students to analyze certain study characteristics (i.e., learning demands, study stressors, and study resources) was used in this research to gain insights into whether and how these characteristics have changed during the pandemic and how they relate to students’ well-being and mental health.

## 2. Methods

### 2.1. Study Design

The data were collected within the Well-Med project (partly funded by the Austrian Science Fund, P27228-G22). This project is a cross-sectional and longitudinal (over a maximum period of 6 years—duration of medical studies) cohort study which aimed to investigate the various individual and situational factors influencing the well-being and health of medical students and hospital physicians.

### 2.2. Setting and Context

With institutional review board approval, all the medical students (human medicine and dentistry) were invited via mail to complete an online survey [55]. The purpose of the study was explained in the email, including the link to the online survey. The invitation link was sent out by the IT department of the university, and anonymous data transfer was guaranteed through the use of an encrypted server. A token system was used to ensure anonymity and allow longitudinal re-identification of the participating students. Incentives comprised medical education credits and direct automated feedback on their survey results. The survey was always conducted once halfway through the academic year, between January and February/March. The data collected between 2017 and 2020 were categorized as “pre-pandemic”, reflecting regular medical study conditions, and the data from 2022 were labelled as “peri-pandemic”, reflecting two years of various COVID measures during the medical study period. The data from 2020 were collected before the first “strict lockdown” in Austria started, which was also considered the cut-off date for being “pre-pandemic” in this study. When the data were collected in March 2022, some courses were held online (e.g., problem-oriented learning). Lectures were streamed or could be attended in person (with mandatory use of FFP2 masks) depending on the number of participating students, and practical courses and high-stake examinations were held in person considering all valid measures (mandatory FFP2 masks and verification of being either (1) vaccinated at least twice, (2) recently recovered from COVID-19 (within the last 6 months), or (3) PCR-tested as COVID-19-negative within the last 3 days). Due to the unprecedented circumstances during the pandemic, no data were collected in 2021.

### 2.3. Participants and Sample Size

All active medical students at the university were eligible to participate in the study and were invited. Therefore, it would have been possible to collect a maximum of 2400 data sets per year (400 students per academic year over the 6-year study duration). 

### 2.4. Variables and Measurements

#### 2.4.1. Mental Health

Until 2020, the German version of the “Short Form Health Survey” (SF-12) including 12 questions was used to assess the participants’ mental health (in addition to physical health) in the past four weeks [56]. The standardized Mental Health Composite Score (MCS) results from all 12 items (raw data ranging from 0 to 100), which need to be weighted using various regression coefficients and summed. The MCS is presented as a T-score (mean 50 ± 10), with higher scores indicating better mental health. The SF-12 reliability is very difficult to calculate due to the specific characteristics of the scale, since it requires reliability estimates for each of the underlying subscales, some of which only have one item. Moreover, the different weighting of each individual item in relation to the total scale and the different response formats of the subscales are problematic [57]. Therefore, reference data from another study examining medical students’ mental health will be adopted with an internal consistency of α = .84 [58]. One item example is “How much of the time did you feel discouraged or depressed?”

In 2022, parts of the “Patient Health Questionnaire” replaced the SF-12 to assess mental health, including the nine-item depressive symptom severity scale (PHQ-9) and the seven-item generalized anxiety disorder scale (GAD-7) to screen, identify, and measure questionnaires based on the diagnostic criteria from the “Diagnostic and Statistical Manual of Mental Disorders, 4th edition” (DSM-IV). The responses to the PHQ-9 and GAD-7 items are given on a four-point Likert scale ranging from “not at all” (=0) to “nearly every day” (=3). The recommended cut-off scores for identifying probable cases of moderate depression and generalized anxiety disorder were used (PHQ-9: cut-off ≥ 10 points with 88% sensitivity and 88% specificity; GAD-7: cut-off ≥ 8 points with 92% sensitivity and 76% specificity) [59,60,61]. The Cronbach’s alpha values in this sample were good for the PHQ-9 (α cross-sectional = .83; α peri = .81) and the GAD-7 (α cross-sectional = .85; α peri = .88). Item examples include “Over the last 2 weeks, how often have you been bothered by the following problems: e.g., feeling nervous, anxious or on edge/trouble relaxing” (GAD-7) and “little interest or pleasure in doing things/feeling down, depressed, or hopeless” (PHQ-9).

#### 2.4.2. Thriving 

The validated German “Comprehensive Inventory of Thriving” (CIT) was used to measure well-being [62]. The questionnaire includes 18 subscales with 54 items, which can be answered on a five-point scale ranging from “strongly disagree” (=1) to “strongly agree” (=5), covering a broad range of well-being components. Three subscales were assigned to SWB and fifteen to PWB as composite scores. SWB comprises life satisfaction and positive and negative emotions, while PWB includes autonomy, engagement, meaning, mastery (accomplishment, learning, self-efficacy, self-worth, and skills), optimism, and relationships (belonging, community, loneliness, respect, support, trust). Each subscale is measured with three items. An overall mean well-being value was also calculated (= thriving), with higher scores indicating higher levels of well-being. The Cronbach’s alpha values for SWB (α cross-sectional = .94; α pre = .95; α peri = .93) and PWB (α cross-sectional = .92; α pre = .95; α peri = .92) were excellent in this sample. Item examples included “I feel good most of the time” (SWB; positive emotions), “I can succeed if I put my mind to it” (PWB; self-efficacy), and “I am optimistic about my future” (PWB; optimism).

#### 2.4.3. Burnout

Burnout, including the dimensions of emotional exhaustion (EE), cynicism (CY), and inefficacy (IE), was measured using the 15-item German student version of the “Maslach Burnout Inventory” (MBI-SS-GV) [39]. The responses were given on a 7-point frequency scale from “never” (=0) to “daily” (=6). In the following, only the subscale “emotional exhaustion” as the core symptom of burnout will be included in the analyses. The subscale showed good Cronbach’s alpha values (α cross-sectional = .86; α pre = .83; α peri = .88). Item examples for this dimension include “I feel emotionally drained by my studies” or “I feel used up at university day’s end”.

#### 2.4.4. Study Conditions

The validated German “Work Analysis Measure for Students” (WA-S Screening) was applied to analyze the medical students’ working conditions in an economic and simultaneously theoretically grounded way that is also associated with relevant university-related aspects of well-being and mental health [53]. The 28 items were answered using a five-point Likert scale ranging from “not at all” (=1) to “yes exactly” (=5). The questions were mapped to eight subscales: lecturer feedback, autonomy, participation (these three can be added up to “resources”), organizational stressors, information problems, work overload (these three can be added up to stressors), skill adequacy, and cognitive demands. The cross-sectional Cronbach’s alpha ranged from α = .60 (skill adequacy) to α = .88 (information problems). The pre-pandemic alpha ranged from α = .49 (skill adequacy) to α = .93 (participation), and the peri-pandemic alpha ranged from α = .49 (skill adequacy) to .89 (work overload). Therefore, the subscale “skill adequacy” needs to be interpreted with caution due to the low internal consistency. Item examples include “My lecturers provide clear feedback on my study performance” (lecturer feedback) or “I often have too much work to do at once” (work overload).

#### 2.4.5. Social Support

For analyzing the important resource of social support, which is not included in the measurements of the study conditions (WA-S, see above), the scale of social support from the German “Salutogenetic Subjective Working Analysis” (SALSA) was extracted and adapted into a student version [63]. The global scale, with higher scores indicating higher social support, comprises three questions (someone you can rely on if study problems appear, willingness of someone to listen to your problems, someone who actively supports you to facilitate your study) related to lecturers, study colleagues, and persons outside the university (e.g., partner, friends, family). The responses were given on a five-point scale ranging from “not at all” (=1) to “totally” (=5). In the cross-sectional samples, the Cronbach’s alpha values ranged from α = .74 to .82, the pre-pandemic alpha ranged from .72 to .92, and the peri-pandemic alpha ranged from .70 to .83. 

#### 2.4.6. Study Satisfaction

To measure the satisfaction of the students with their medical studies in general, a single-dimension nine-item scale developed by the German Centre for Higher Education Research and Science Studies was used [64]. The items refer to several aspects of the study and can be answered on a five-point Likert scale ranging from “totally unsatisfied” (=1) to “very satisfied” (=5). In this sample, the Cronbach’s alpha ranged from α = .81 (peri-pandemic) to α = .82 (cross-sectional and pre-pandemic). Item examples include “How satisfied are you with … the lecturer mentoring in your study course?/… the professional quality of your courses?/… the number of participants in your courses?”

#### 2.4.7. COVID-19-Related Questions

Specific questions related to COVID-19 were asked to identify how the impact of the COVID-19 measures on study life was perceived by the medical students, including topics like information about various measures at their university, how they structured their everyday study life, work–life balance, impact of physical distancing, and prevalence of substance use. They were also related to the constructs of thriving and mental health. These questions were self-developed and can be found in Appendix B.

### 2.5. Statistical Methods

All cross-sectional and longitudinal analyses were conducted using the IBM SPSS Statistics 24 software [65]. Only complete data within the respective questionnaires were integrated into the analyses (omitted data are pointed out in the table notes when applicable). To calculate the differences between multiple groups in interval-scaled scales simultaneously (i.e., different study year cohorts from the 1st to the 5th year and measurement time points), an ANOVA was initially used. The Levene test was also used to test for homogeneity of variance. Due to partial lack of variance homogeneity and/or differences in group size, the Kruskal–Wallis test was used as a non-parametric test to calculate group differences instead of an ANOVA (i.e., comparing the study years related to COVID-19-specific questions and checking if any differences exist between the survey years). In order to better understand the results of the Kruskal–Wallis test, they were checked again with a *U* test, whereby the grouping variable was divided into its different dichotomous variations. The differences between dichotomous variables in interval-scaled scales (i.e., comparing pre- and peri-pandemic constructs overall or the sexes) were calculated using *t* tests, either for independent samples in the cross-sectional data or for dependent samples in the longitudinal data. Significance levels with *p* ≤ .01 are marked with two asterisks (**), and *p* ≤ .05 is marked with one asterisk (*). Correlations between interval-scaled variables were calculated using Pearson’s product-moment correlation coefficient, and associations between nominal scaled variables with more than two levels were calculated using Cramer’s *V*. The correlation coefficients can be interpreted as follows: *r* < .10: no correlation; *r* = .10–.29: low correlation; *r* = .30–.49: moderate correlation; and *r* = .50: high correlation [66]. Finally, the effect sizes regarding group differences are represented as Cohen’s *d* (>0.2 = small, >0.5 = medium, >0.8 = large) [66].

If the students participated more than once between 2017 and 2020, their first measurement time point was considered for the cross-sectional comparative analyses (to reflect the medical study in its entirety over different years), whereas their last measurement time point was considered for the longitudinal pre- vs. peri-pandemic analyses (to minimize the impact of other possible influencing factors and confounders) to avoid measurement repetition effects. To adopt this approach, it was first necessary to consider in a previous step whether the data from the years 2017–2020 prior to the pandemic significantly differed from each other. Since the Levene test did not confirm variance homogeneity for all individual scales and a huge difference in sample size between the years became apparent, the Kruskal–Wallis test for interval-scaled data and Cramer’s *V* for nominal-scaled data were used instead of a one-way ANOVA as non-parametric procedures to calculate if any differences existed between the survey years. None of the variables included in the further analyses differed significantly between the pre-pandemic years of 2017 to 2020, justifying the data aggregation. 

## 3. Results

The sociodemographic data for the cross-sectional as well as longitudinal design are presented in Table 1. 

### 3.1. Cross-Sectional Analyses

A total of 988 participants were included in these analyses. Most of them were female (*N* = 571, 57.8%), with a mean age of 21.78 (*SD* = 2.63). The average study semester of the participants was 3.49 (*SD* = 2.41). More than half of the students (*N* = 530; 53.6%) were Austrian citizens, lived in shared apartments (*N* = 511, 51.7%), and were single (*N* = 527, 53.3%). A total of 328 students reported having a job alongside their study (33.2%; mean weekly working hours = 10.76, *SD* = 7.48), and 45 students (9.4%) reported that they suffered from financial problems due to the COVID-19 environment.

Table 2 shows the intercorrelations between the respective scales (comprising thriving, mental health, burnout, study satisfaction, study conditions, and social support). The SF-12 Mental Health Composite Score (MCS) that was used up to and including the year 2020 (*N* = 480) revealed a mean *T*-value of 41.48 (*SD* = 11.96) within the normal range (*T*-values from 40 to 60); however, this is rather low. Additionally, 42.9% of the students had a *T*-value below (<40) the normal range, and only 1.2% of the students had a *T*-value above the normal range (>60). When analyzing the GAD and PHQ scores for symptoms of anxiety (*N* = 423; *M* = 6.24; *SD* = 4.21) and depression (*N* = 419; *M* = 6.38; *SD* = 4.61) collected in 2022 as well as the recommended cut-off values, in total, 140 students (14.2%) fulfilled the screening criteria for generalized anxiety disorder and 97 students (9.8%) fulfilled those for depression. 

Females had significant higher scores on both the GAD (*N*_female_ = 250; *N*_male_ = 164) and the PHQ (*N*_female_ =249; *N*_male_ = 163) than males (GAD: *M* = 7.01, *SD* = 4.11 vs. *M* = 4.91, *SD* = 3.96; *T* = 5.14, *p* < .001, *d* = 0.52; PHQ: *M* = 7.00, *SD* = 4.65 vs. *M* = 5.20, *SD* = 4.18; *T* = 4.01, *p* < .001, *d* = 0.40), but not the MCS (females: *M* = 39.75, *SD* = 11.79; males: *M* = 44.29, *SD* = 11.73). Looking at the normal range of the MCS, 49.2% of women and 32.8% of men were below this range, while 2.7% of men and 0.3% of women were above. Approximately 17% of women compared to 10.4% of men fulfilled the screening criteria for generalized anxiety disorder, and 11.4% of women compared to 7.6% of men fulfilled the screening criteria for depression. 

The items of the SF-12 revealed that 41.7% of the students were mostly “calm and peaceful” (vs. 22.1% who answered rarely), 40.2% sometimes had a “lot of energy” (vs. 21.5% who answered rarely), and 42.1% were sometimes “downhearted and blue” in the last four weeks (vs. 14.8% who answered mostly). In the GAD, 55.3% indicated that they were anxious or nervous and were not able to control their thoughts (42.4%) on some days during the last two weeks. Approximately 14.9% had problems relaxing on more than half of the days, while 35.3% feared that something bad could happen on some days in the last two weeks. The PHQ revealed that 50.6% had, on some days in the last two weeks, “little interest or pleasure in activities” (vs. 9.1% on more than half of the days). Approximately 10.3% reported feeling “depressed, melancholic or hopeless” and were “tired or without energy” (22.0%) on more than half of the days. It is also notable that, in total, 10% stated that they would have preferred “to be dead or to cause themselves harm” on some days in the last two weeks.

Then, by applying the dichotomous pre- vs. peri-pandemic variable, significant differences were found in various scales, all indicating a higher burden in the pre-pandemic group concerning the burnout subscale “emotional exhaustion” (*d* = 0.27), social support in total (*d* = 0.17) and particularly relating to lecturers (*d* = 0.29), and study satisfaction (*d* = 0.24), as well as the study condition subscales organizational stressors (*d* = 0.40), cognitive demands (*d* = 0.23), work overload (*d* = 0.52), and information problems (*d* = 0.26; see Table 3). When repeating the analysis separately for each sex, additional differences in PWB (*d* = 0.21), thriving (*d* = 0.19), and information problems (*d* = 0.32) were found for women. For men, further differences in participation (*d* = 0.28) and study satisfaction (*d* = 0.34) became apparent. This analysis was repeated again, now investigating the study years individually (except for the 6th year, as there was only one person in the respective pre-pandemic group) to examine whether the results were influenced by the study, as the different study years contain different emphases and workloads. Overall, the students in their 2nd study year before the pandemic showed the largest differences compared to the 2nd-year students during the pandemic, with the latter reporting fewer burnout symptoms and stressors, more social support from lecturers, and study satisfaction. All the significant results can be found in Table 3, whereas a comprehensive overview including the non-significant results can be found in Appendix A.

Finally, the responses to selected COVID-19-specific questions from the peri-pandemic group were analyzed (questions listed in Appendix B). A total of 161 students (31.6%) suffered from a COVID-19 infection in the last 12 months. Most of the students reported that they have consumed less or at most equal amounts of alcohol (85.3%) and drugs (89.8%) compared to the pre-pandemic period; nevertheless, 8.3% reported drinking (much) more alcohol and 3.7% used (much) higher amounts of drugs (see Table 1). Approximately 9.6% experienced strong effects of the pandemic containment measures on their studies, with 23% mostly or fully agreeing that they were struggling with organizing their everyday study time, and 33% even stated that during online teaching, they had difficulties in distancing themselves from their studies at home or during their free time. Approximately 30.1% mostly or totally disagreed that they were able to maintain personal contact with fellow students despite not physically attending lectures, although 45.2% mostly or fully agreed that they were burdened by the lack of physical attendance. More than half of the students (57.7%) mostly or fully agreed that they had difficulties making new friends due to the lack of or limited attendance. However, an overwhelming majority (80.1%) agreed, mostly or completely, that they felt sufficiently informed about the COVID-19 prevention measures at their university.

Relating the COVID-19-specific questions to the measured constructs (see Table 4) indicated that experiencing a COVID-19 infection did not affect the students’ overall ability to thrive. However, students reporting difficulties in trying to “...organize everyday study time in a meaningful way” or “distance oneself from studies at home or during free time” also reported lower levels of well-being and more symptoms of anxiety, depression, and emotional exhaustion (*r* = .14 to −.23). In contrast, personal contact with fellow students and sufficient information about the COVID-19 prevention measures at their university were significantly related to higher well-being (*r* = .13 to .17). However, feeling sufficiently informed was not related to anxiety or depression symptoms. 

The COVID-19-specific questions were further analyzed to examine whether there were differences between the different study years (see Table 5; a comprehensive overview including non-significant results can be found in Appendix A). Significant differences were found concerning: “How much impact do the pandemic containment measures have on your student life?” (question IV), “I feel sufficiently informed about the Corona prevention measures at my university” (question V), and “Due to the lack of or limited attendance during my studies, I find it difficult to make new friends” (question X). With regard to how much impact the pandemic containment measures had on student life, (a) students in the 1st year reported a significantly higher impact on their life than students in their 2nd, 3rd, and 4th year of studies. Students in their 1st year felt significantly more informed than 3rd or 5th year students (b), and (c) they also perceived less difficulties making new friends despite the lack of or limited attendance during their studies than 2nd, 4th, and 5th year students.

### 3.2. Longitudinal Analyses

For the longitudinal dataset, only the students who participated in the surveys from 2017, 2018, 2019, or 2020 and in 2022 were included (*N* = 63). These 63 individuals (please see sociodemographic data in Table 1) all answered the survey in the year 2022, with 1 of these students participating in 2017 for the last time, 10 in 2018, 21 in 2019, and 31 students in 2020. A total of 41 students (65.1%) from 2022 were female and, on average, in their eighth semester (*SD* = 1.89). The mean age was 23.87 years (*SD* = 2.13), and 37 participants (58.7%) were from Austria. Almost half of the students (*N* = 29; 46%) lived in shared apartments, and 38 students were in a relationship (60.3%). A total of 20 students reported having a job alongside their studies (31.7%; mean weekly working hours = 9.33, *SD* = 4.75), and 8 students (12.7%) reported that they suffered from financial problems due to the COVID-19 environment.

In the pre-pandemic cohort, overall differences were found concerning higher emotional exhaustion and cognitive demands, more stressors (organizational stressors, work overload, information problems), and less social support and study satisfaction. Therefore, the longitudinal data were examined to determine whether the same effects could be observed over time on an individual level (differences between the last survey completed before the pandemic vs. the same person’s scores during the pandemic in 2022). Significant differences were found in the subscales of “work overload” (*N* = 32, *T* = 3.15, *p* = .004, *d* = 0.56) and “study satisfaction” (*N* = 50, *T* = 2.65, *p* = .011, *d* = 0.37). In terms of work overload, the students felt more overloaded before the pandemic (*M* = 3.35, *SD* = 1.0 vs. *M* = 2.77, *SD* = 1.08); however, on an individual level, they were now less satisfied with their studies during the pandemic (*M* = 3.77, *SD* = 0.62 vs. *M* = 3.5, *SD* = 0.62), contrary to the cross-sectional results (*d* = 0.24). An overview of all the results can be found in the Appendix A.

## 4. Discussion

Even before the COVID-19 pandemic beginning in March 2020, medical students around the world reported high levels of demands and stressors accompanied by a lack of resources, therefore experiencing more burnout symptoms and mental strain (anxiety, depression, loneliness, alcohol/substance abuse, and suicidal ideation) compared to an age-matched general population. In combination with quickly adapting study conditions due to the pandemic, which entailed new challenges (e.g., digital learning/teaching, physical distancing, organizing study time, finding a new work–life balance), existing studies reported a strong impairment of well-being in almost three quarters of students [38] or high prevalence rates for emotional exhaustion, with up to almost every second student being affected [47]. 

In the present analysis, unexpectedly, no deterioration of thriving was found when comparing the pre- and peri-pandemic data. Interestingly, the students (particularly in their 2nd study year) included in the cross-sectional comparisons reported that they “felt better” during the pandemic compared to before in terms of more social support from lecturers, more study satisfaction, and less emotional exhaustion and stressors (organizational stressors, cognitive demands, work overload, information deficit). In the longitudinal design with matched data on an individual level, work overload was also rated higher before the pandemic. In contrast, less study satisfaction was reported during the pandemic. Work overload includes consistently working faster but still not completing all tasks, recurrent work peaks, time pressure due to tight deadlines, and low quality of work due to the huge workload. Regarding work overload, the advantages of digital teaching could include higher flexibility, learning at one’s own speed, no overlapping lectures, and time savings for either more learning or leisure time [22], with “e-learning satisfaction” as a specific aspect of study satisfaction [67]. Moreover, later study years in medicine comprise many more diverse practical/clinical courses with fewer semester hours than in the beginning, with more theoretical and quite comprehensive courses (anatomy, chemistry, histology, physics, and physiology). Therefore, the workload itself may be generally evaluated as higher by the students in the beginning, or they may be better able to cope with it later on. In terms of overall study satisfaction, the cross-sectional and longitudinal data seem to be more complex. The different sample sizes here might have played a role, or there is a possibility that the students who were less satisfied participated more often in the survey to express their opinion, which could explain the lower values in the long term. 

The lower peri-pandemic study satisfaction in the longitudinal data could be also explained by prevailing pandemic-related circumstances. Typically, these included mandatory FFP2 mask wearing when attending lectures; maintaining physical distancing; continuously providing evidence of being vaccinated, being recently recovered, or negatively PCR-tested; and a mixture of in-person and digital teaching, implying more effort to coordinate practical courses at the university vs. online courses. Furthermore, the largest portion of students in the longitudinal design were those who were exposed to the most study changes due to the pandemic, including their memories of “former normal conditions and proceedings” as they started their study before the pandemic in autumn 2019. This may have also led to the feeling of being “demoralized” or feeling less well-educated compared to other medical students who started earlier without experiencing the pandemic to its fullest extent or those who started later. In particular, they felt less satisfied in terms of lecturers’ support, teaching, and professional quality, and the content of different subjects, study structure, number of participants in courses, educational supplies, and equipment, study department, and learned knowledge and skills. 

On a general level, the main explanation for the constant high thriving results and higher satisfaction during the pandemic is a possible underlying response-shift phenomenon [68]. Response-shift effects refer to changes in the meaning of one’s self-evaluation of a target construct, e.g., thriving, as a result of (1) a change in the person’s internal standards of measurement (recalibration), (2) a change in the person’s values (reprioritization), or (3) a redefinition of the target construct by either adding completely new values or removing “outdated” values, as they have lost their former meaning (reconceptualization). For an easier understanding, scale recalibration can be compared with one’s pain perception on a 10-point Likert scale (0 = no pain, 10 = max. imaginable pain). For example, in 2020, a person suffers from a knee injury, and when being asked how strong the pain is on this scale, the person answers 8. In 2021, the same person has kidney stones/colic, and when being asked then how strong the pain is (0–10), the person answers 8 again. Looking back from now, the person would have rated the pain of the knee injury with 4, reflecting a scale recalibration [69]. The same could have happened in terms of a “thriving recalibration”—in 2020, medical students perceived a medium well-being and rated it a 3 on a 1- to 5-point scale. But in 2022, after two years of living/studying during the pandemic, they feel “ok” in terms of thriving and rate their experience as a 3 again; however, when looking back from now, they would rate their thriving as a 5 (=high value) in 2020. Thus, the students’ anchor points may have shifted downwards. Therefore, although they may have actually experienced a lower well-being during the pandemic, this was not apparent in the responses to the questionnaires. Reprioritization describes changes in relation to the meaningfulness of one’s personal values. Here, reprioritization effects could be existent in terms of higher study satisfaction in the cross-sectional analyses. Due to the modified peri-pandemic living conditions, the medical studies rose in personal value to such an extent (students are really happy to study at all despite the pandemic) that the students felt more satisfaction. 

These response-shift effects could also possibly explain the results concerning greater perceived social support from lecturers, less emotional exhaustion, and fewer stressors (organizational stressors, cognitive demands, information deficit) during the pandemic. Another explanation might be that there were lower study requirements during the 2022 period when the data were collected compared to the pre-pandemic times, as there was still a wide array of digital and in-person teaching modalities with optional choices of how to attend, making it easier for individuals to organize themselves. In addition, the university and its lecturers may have also implicitly granted more lenience after two pandemic years. Furthermore, the pandemic measures were already clearly reduced in spring 2022, heralding the feeling of an end to the pandemic and potentially generating a feeling of relief in the medical students, which was reflected in the “better” results. Finally, one could also assume that, overall, the pandemic measures at the medical universities were not as restrictive as at other Austrian universities (complete closure of the studies or cancellation of in-person courses over a longer time period, changing exclusively to digital teaching/learning), thus explaining the peri-pandemic results. 

Cross-sectionally, the first-year students felt better informed about the prevention measures at their university than students did in later years. Presumably (and comprehensively), the university turned its attention to the new students in particular, with all relevant information disseminated in the main entrance course before the semester starts, while more knowledge or individual responsibility was attributed to the other students. The students in later study years also perceived more limitations and difficulties in terms of socializing, as they possibly compared it with the pre-pandemic conditions and processes when it was “easier” (different), while new students did not experience the former study atmosphere.

The number of students reporting anxious and depressive symptoms above the cut-off value for possible clinical cases was rather low compared to other studies reporting peri-pandemic prevalence rates. Approximately every seventh student fulfilled the screening criteria for possible cases of generalized anxiety disorder (14.2%), and approximately every tenth student exceeded the cut-off value for major depression (9.8%). The observed increases during the pandemic (up to 29% for anxiety and 35% for depressive symptoms) exceeding the clinically relevant cut-off in Austrian students in other studies [27] were not present in our data. This could be explained by the time frame when the data were collected and the reduced pandemic measures, or by the comparatively fewer cancellations of lectures at the medical universities compared to other Austrian universities. Other international studies underline this assumption that the results are dependent on the particular period when the data were collected, e.g., May/June 2020 [70,71] or March 2021 [72], and the respective ongoing measures seem to be crucial in revealing increased rates of anxiety and depressive symptoms in medical students. Nonetheless, when associating the peri-pandemic GAD and PHQ data with the pre-pandemic MCS, where 22.1% rarely felt “calm and peaceful” and 14.8% mostly felt “downhearted and blue” in the last four weeks, one can assume quite stable values in terms of mental health, although the comparison is limited. Caution should be taken when interpreting these results, since showing anxious or depressive behavior does not necessarily mean that it is emanating from a clinical disorder, as it may be from heavy stress, sleep deprivation, or any other “non-clinical” burden, particularly in light of the demands of medical studies. Overall, the percentages were comparable to those before the pandemic, and no serious deterioration was detected. 

Experiencing a COVID-19 infection in the last 12 months was not significantly correlated with the students’ perceived well-being, suggesting that the medical students from this sample endured the infection quite well in terms of thriving. Possible cases or consequences of a post-COVID syndrome were not identified. In contrast, students having difficulties in organizing their everyday study time in a meaningful way and in distancing themselves from their studies at home or during their free time during the periods of online teaching experienced significantly lower well-being, as well as more anxiety, depressive symptoms, and emotional exhaustion. These correlations could be an indicator of poor individual structures, but they are also explainable in both directions: (a) due to these difficulties, the students experienced more psychological symptoms, or (b) due to psychological symptoms, the students experienced more difficulties. For any causal assumption, a different study design would have been needed.

### Limitations

There are some limitations of the reported study that need to be addressed. One limitation of this study in terms of biases is the rather low sample size in the longitudinal analyses, and that there were not enough participants to compare each study year separately. The results could have been more comprehensible, as the workloads and study demands vary across the years. Moreover, no peri-pandemic data were collected in 2021 due to the unprecedented circumstances, possibly skewing the results towards a “better” outcome. The large online test battery could be one explanation for the lower participation, as many students did not complete all the questionnaires or did not participate again at later time points. In addition, the incentives comprising education credits and direct automated feedback on their survey results did not trigger as much motivation as was desired. Furthermore, there is a likelihood that more students with a low overall study satisfaction were included in the longitudinal data compared to the students in the cross-sectional data, explaining the inconsistent results. The degree to which these results can be generalized to other (medical) students is limited due to the homogenous sampling (e.g., one culture, the same language, similar working climate, and organizational structures at one medical university).

## 5. Conclusions

Overall, the medical students in this Austrian sample reported higher levels of social support and study satisfaction and less emotional exhaustion, cognitive demands, and stressors during the pandemic compared to before the pandemic. On a general level, the students maintained a constant high level of well-being during the pandemic. Possible response-shift effects which might have occurred need further exploration. Finally, it has not yet become the standard to draw attention to (medical) students’ well-being and enhance their “thriving”. The focus of research involving medical students tends to be on pathology (as medicine itself has a strong orientation towards pathology) and is mainly addressed only when problems have already occurred, such as failed exams or mental health diagnoses. A consensus definition on well-being, combined with a gold standard tool to capture well-being in different populations, would support endeavors to address and enhance medical students’ well-being.

## Figures and Tables

**Table 1 behavsci-14-00007-t001:** Sociodemographic data.

Variable	Cross-Sectional Data		Longitudinal Data			
			Pre-Pandemic		Peri-Pandemic	
	Sample Size (*N*)	%	Sample Size (*N*)	%	Sample Size (*N*)	%
Survey time						
2017	147	14.9	1	1.6	-	-
2018	169	17.1	10	15.9	-	-
2019	126	12.8	21	33.3	-	-
2020	100	1.1	31	49.2	-	-
2022	446	45.1	-	-	63	100
Sex						
Male	356	36.0	19	30.2	20	31.7
Female	571	57.8	39	61.9	41	65.1
Nationality						
Austrian	530	53.6	33	52.4	37	58.7
German	209	21.2	8	12.7	8	12.7
Italian	155	15.7	16	25.4	15	23.8
Other	32	3.2	1	1.6	1	1.6
Living situation						
Alone	192	19.4	10	15.9	7	11.1
With partner	84	8.5	5	7.9	13	20.6
In shared flat	511	51.7	32	50.8	29	46.0
With family	139	14.1	11	17.5	12	19.0
Family status						
Single	527	53.3	36	57.1	21	33.3
Married	8	.8	0	0	2	3.2
Relationship	384	38.9	22	34.9	38	60.3
Other	7	.7	0	0	0	0
Study semester						
First	451	45.6	30	47.6	0	0
Second	250	25.3	5	7.9	0	0
Third	149	15.1	9	14.3	0	0
Fourth	90	9.1	0	0	2	3.2
Fifth	34	3.4	9	14.3	0	0
Sixth	14	1.4	0	0	15	23.8
Seventh	0	0	3	4.8	0	0
Eighth	0	0	2	3.2	12	19.0
Ninth	0	0	0	0	0	0
Tenth	0	0	0	0	19	30.2
Have a side job						
Yes	328	33.2	17	27	20	31.7
No	598	60.5	41	65.1	41	65.1
COVID-19 infection (in last 12 months)						
Yes	161	31.6	-	-	20	31.7
No	317	62.3	-	-	42	66.7
Alcohol in comparison *						
Much less	93	18.3	-	-	9	14.3
Less	149	29.3	-	-	21	33.3
Equal	192	37.7	-	-	27	42.9
More	37	7.3	-	-	5	7.9
Much more	5	1.0	-	-	0	0
Other drugs in comparison *						
Much less	89	17.5	-	-	4	6.3
Less	33	6.5	-	-	4	6.3
Equal	335	65.8	-	-	50	79.4
More	175	3.3	-	-	4	6.3
Much more	2	.4	-	-	0	0

Note: Data missing if sample sizes do not equal N or 100% for each group; * compared to before the pandemic.

**Table 2 behavsci-14-00007-t002:** Intercorrelations.

	Age	Sex	PWB	SWB	EE	SA	CD	FBL	AUT	PT	OS	WO	IP	SSL	SSC	SSP	SST	GAD	PHQ	CIT	SS
Age	1																				
Sex	.12 **	1																			
PWB	−.14 **	.04	1																		
SWB	−.10 **	.05	.74 **	1																	
EE	.07	−.01	−.34 **	−.39 **	1																
SA	−.06	.08 *	.47 **	.37 **	−.36 **	1															
CD	−.07	−.08 *	.18 **	.03	.18 **	.05	1														
FBL	−.04	.07	.21 **	.18 **	−.18 **	.18 **	.08 *	1													
AUT	−.16 **	.02	.17 **	.15 **	−.23 **	.18 **	.08 *	.07 *	1												
PT	.01	.05	.05	.07	−.17 **	.10 **	.00	.39 **	.15 **	1											
OS	.18 **	.04	−.17 **	−.19 **	.41 **	−.21 **	.17 **	−.11 **	−.19 **	−.01	1										
WO	.07 *	−.06	−.13 **	−.21 **	.56 **	−.22 **	.31 **	−.14 **	−.17 **	−.15 **	.49 **	1									
IP	.12 **	.01	−.19 **	−.22 **	.39 **	−.22 **	.07	−.17 **	−.17 **	−.09 **	.62 **	.44 **	1								
SSL	−.04	.05	.21 **	.23 **	−.31 **	.21 **	.03	.48 **	.13 **	.35 **	−.29 **	−.28 **	−.32 **	1							
SSC	−.15 **	−.03	.40 **	.31 **	−.19 **	.23 **	.07	.09 *	.12 **	−.01	−.12 **	−.08 *	−.16 **	.20 **	1						
SSP	−.01	−.10 **	.39 **	.28 **	−.07	.16 **	.14 **	.07	.06	−.01	−.04	−.01	−.09 *	.05	.31 **	1					
SST	−.12 **	.02	.38 **	.34 **	−.33 **	.28 **	.06	.39 **	.16 **	.24 **	−.28 **	−.24 **	−.32 **	.82 **	.73 **	.21 **	1				
GAD	.00	−.25 **	−.37 **	−.53 **	.39 **	−.22 **	.11 *	−.04	−.19 **	−.06	.12 *	.25 **	.16 **	−.18 **	−.15 **	−.05	−.21 **	1			
PHQ	.04	−.19 **	−.49 **	−.62 **	.46 **	−.24 **	.04	−.05	−.17 **	−.03	.17 **	.24 **	.20 **	−.17 **	−.18 **	−.14 **	−.22 **	.71 **	1		
CIT	−.14 **	.05	.98 **	.85 **	−.37 **	.46 **	.15 **	.21 **	.17 **	.06	−.18 **	−.16 **	−.21 **	.29 **	.40 **	.38 **	.39 **	−.43 **	−.55 **	1	
SS	−.14 **	.02	.36 **	.34 **	−.41 **	.31 **	.08 *	.31 **	.24 **	.19 **	−.38 **	−.33 **	−.49 **	.45 **	.16 **	.10 **	.41 **	−.22 **	−.29 **	.37 **	1

*N* varies between 360 and 927. ** *p* < .01. * *p* < .05. PWB = psychological well-being, SWB = subjective well-being, EE = emotional exhaustion, SA = skill adequacy, CD = cognitive demands, FBL = feedback of lecturers, AUT = autonomy, PT = participation, OS = organizational stressors, WO = work overload, IP = information problems, SSL = social support from lecturers, SSC = social support from colleagues, SSP = social support private, SST = social support total, GAD = anxiety, PHQ = depression, CIT = thriving, SS = study satisfaction.

**Table 3 behavsci-14-00007-t003:** Differences in the cross-sectional sample.

	Variable	Sample Size Pre (*N*)	Sample Size Peri (*N*)	*T*-Value	*p*-Value	Mean ± SD Pre	Mean ± SD Peri	Scale Range
Overall	EE	384	360	3.62	<.001	2.87 ± 1.09	2.59 ± 1.02	0–6
	OS	400	369	5.60	<.001	2.92 ± 0.82	2.60 ± 0.78	1–5
	CD	400	369	3.15	.002	4.10 ± 0.74	3.93 ± 0.75	1–5
	WO	400	369	7.17	<.001	3.37 ± 0.95	2.90 ± 0.87	1–5
	IP	400	369	3.66	<.001	2.92 ± 0.99	2.66 ± 0.94	1–5
	SST	400	368	2.31	.021	3.35 ± 0.57	3.45 ± 0.61	1–5
	SSL	400	368	3.97	<.001	2.54 ± 0.84	2.77 ± 0.80	1–5
	SS	492	360	−3.44	<.001	3.66 ± 0.59	3.81 ± 0.61	1–5
Men	PT	162	148	−2.41	.016	1.73 ± 0.81	1.96 ± 0.85	1–5
	SS	187	144	−3.03	.003	3.65 ± 0.64	3.86 ± 0.63	1–5
Women	PWB	217	227	2.21	.028	3.95 ± 0.42	3.86 ± 0.42	1–5
	CIT	217	227	2.01	.045	3.94 ± 0.45	3.85 ± 0.45	1–5
	IP	238	221	3.42	.001	2.93 ± 0.98	2.63 ± 0.93	1–5
1st year	EE	220	124	2.19	.029	2.75 ± 1.04	2.50 ± 1.01	0–6
	OS	231	127	3.47	.001	2.76 ± 0.80	2.46 ± 0.74	1–5
	WO	231	127	3.22	.001	3.29 ± 0.93	2.99 ± 0.78	1–5
	SSL	231	127	−2.83	.005	2.66 ± 0.79	2.91 ± 0.84	1–5
2nd year	EE	76	121	2.11	.036	3.02 ± 1.15	2.71 ± 0.89	0–6
	CD	78	121	3.04	.003	4.22 ± 0.66	3.90 ± 0.77	1–5
	AUT	78	121	−2.21	.029	3.49 ± 0.94	3.77 ± 0.78	1–5
	PT	78	121	−2.76	.006	1.54 ± 0.78	1.85 ± 0.75	1–5
	OS	78	121	4.82	<.001	3.20 ± 0.74	2.66 ± 0.79	1–5
	WO	78	121	5.70	<.001	3.62 ± 0.91	2.88 ± 0.88	1–5
	IP	78	121	3.46	.001	3.15 ± 0.85	2.69 ± 0.95	1–5
	SSL	78	121	−3.78	<.001	2.32 ± 0.88	2.77 ± 0.80	1–5
	SS	98	121	−3.99	<.001	3.53 ± 0.54	3.82 ± 0.55	1–5
3rd year	WO	41	66	3.35	.001	3.48 ± 0.81	2.86 ± 0.97	1–5
	SSC	41	65	−2.20	.030	3.94 ± 0.74	4.24 ± 0.62	1–5
	SS	57	62	−2.45	.016	3.49 ± 0.55	3.75 ± 0.62	1–5
4th year	EE	33	35	3.07	.003	3.22 ± 1.17	2.38 ± 1.08	0–6
	OS	35	36	3.62	.001	3.29 ± 0.85	2.56 ± 0.86	1–5
	WO	35	36	2.44	.017	3.32 ± 1.07	2.75 ± 0.90	1–5
	IP	35	36	2.35	.021	3.19 ± 1.02	2.64 ± 0.96	1–5
5th year	IP	14	12	2.75	.012	3.43 ± 1.02	2.56 ± 0.56	1–5

Legend: PWB = psychological well-being, EE = emotional exhaustion, CD = cognitive demands, AUT = autonomy, PT = participation, OS = organizational stressors, WO = work overload, IP = information problems, SSL = social support from lecturers, SSC = social support from colleagues, SST = total social support, CIT = thriving, SS = study satisfaction.

**Table 4 behavsci-14-00007-t004:** Correlations between well-being, mental health, and emotional exhaustion with COVID-19-specific questions.

Questions				Correlations				
			CIT	PWB	SWB	GAD	PHQ	EE
(I)	COVID-19 infection	*r*	−.05	−.06	−.03	−.04	.04	.03
(II)	COVID-19 and alcohol	*r*	−.03	−.03	−.04	.06	.12 *	.07
(III)	COVID-19 and intoxicants	*r*	−.03	−.05	.01	.03	.08	−.03
(IV)	Impact of measures	*r*	−.04	−.04	−.06	.10 *	.08	.13 **
(V)	Informed about measures	*r*	.13 **	.14 **	.07	−.08	−.07	−.21 **
(VI)	Organize study time	*r*	−.23 **	−.22 **	−.20 **	.17 **	.23 **	.16 **
(VII)	Distancing from studies	*r*	−.19 **	−.17 **	−.20 **	.20 **	.18 **	.43 **
(VIII)	Personal contact with colleagues	*r*	.17 **	.15 **	.17 **	−.13 **	−.11 *	−.07
(IX)	Physical distancing	*r*	−.07	−.06	−.09	.09	.14 **	.05
(X)	Difficulties making new friends	*r*	−.20 **	−.18 **	−.19 **	.10 *	.13 **	.06

Sample size varies between *N* = 416 and *N* = 478. * *p* < .05. ** *p* < .01. CIT = thriving, PWB = psychological well-being, SWB = subjective well-being GAD = anxiety, PHQ = depression, EE = emotional exhaustion.

**Table 5 behavsci-14-00007-t005:** Differences between study years in the responses to COVID-19-specific questions.

Question	Compared Study Years	Sample Size	Middle Ranks	*U*-Value	*p*-Value
(IV) How much impact do the pandemic containment measures have on your student life?	1 vs. 2	1:147	12.46	6829.50	<.001
		2:130	159.97		
	1 vs. 3	1:147	107.46	4918.00	<.001
		3:91	138.96		
	1 vs. 4	1:147	91.88	2628.00	<.001
		4:53	124.42		
(V) I feel sufficiently informed about the Corona prevention measures at my university.	1 vs. 3	1:146	129.36	513.50	.001
		3:91	102.38		
	1 vs. 5	1:146	94.07	1961.50	.034
		5:34	75.19		
(X) Due to the lack of or limited attendance during my studies. I find it difficult to make new friends.	1 vs. 2	1:142	125.54	7673.00	.012
		2:130	148.48		
	1 vs. 4	1:142	91.04	2774.00	.006
		4:52	115.15		
	1 vs. 5	1:142	82.60	1575.50	.001
		5:34	113.16		

## Data Availability

The data presented in this study are available from the corresponding authors upon request.

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
