# Peer review of "Well-Being, Mental Health, and Study Characteristics of Medical Students before and during the Pandemic"

_behavsci, 2023, doi:10.3390/bs14010007_

Round 1

Reviewer 1 Report

Comments and Suggestions for Authors

Dear authors,
Thank you very much for the opportunity to review this review. I read the work with great interest. It carries a lot of useful information, however, before making a decision, many inaccuracies should be clarified:

Introduction :

The authors could provide a more detailed description of the organisation of medical studies in Austria, specifying whether there are differences in the distribution of theoretical and practical learning over the years related to workload.

Materials and Methods :

The description of the text P5 l182 appears different from the presentation of the results in the table. The authors should correct this anomaly

Results :

 In Table 1, what is the reference period for changes in alcohol and drug consumption?

 Table 4: Why is there no  presentation of all the categories? ( 1 vs 2, 1 vs3, 1 vs4, 1 vs5)

Discussion :

The authors could draw on other recent articles to discuss their results

-        Chumakov, E., Petrova, N., Mamatkhodjaeva, T., Ventriglio, A., & Bhugra, D. (2022). The impact of covid-19: Anxiety, depression, and wellbeing among medical students. International Journal of Social Psychiatry, 68(6), 1270-1276.

-        Avila-Carrasco, L., Díaz-Avila, D. L., Reyes-López, A., Monarrez-Espino, J., Garza-Veloz, I., Velasco-Elizondo, P., ... & Martinez-Fierro, M. L. (2023). Anxiety, depression, and academic stress among medical students during the COVID-19 pandemic. Frontiers in Psychology, 13, 1066673.

-        AlJhani, S., Alateeq, D., Alwabili, A., & Alamro, A. (2022). Mental health and online learning among medical students during the COVID-19 pandemic: a Saudi national study. The Journal of Mental Health Training, Education and Practice, 17(4), 323-334.

Author Response

Comments and Suggestions for Authors

Dear authors,
Thank you very much for the opportunity to review this review. I read the work with great interest. It carries a lot of useful information. However, before making a decision, many inaccuracies should be clarified:

Thank you very much for your valuable review. Please find our answers and revisions made in the paper as follows.

Introduction: The authors could provide a more detailed description of the organisation of medical studies in Austria, specifying whether there are differences in the distribution of theoretical and practical learning over the years related to workload.

In the previous version, we already discussed the contents of the medical curriculum in terms of differing workloads (lines 425-428). However, we added some more detailed information in the introduction (lines 45-53).

Materials and Methods: The description of the text P5 L182 appears different from the presentation of the results in the table. The authors should correct this anomaly

This notice is correct – accidentally, we mixed two different logical pathways to describe the longitudinal sample. The numbers in the tables remained the same but we changed the text and the respective numbers accordingly (lines 390-391).

Results: In Table 1, what is the reference period for changes in alcohol and drug consumption?

The reference period is before the beginning of the pandemic. Please see Appendix with the listed items (II & III), however, we added the information to table 1.

Table 4: Why is there no presentation of all the categories? (1vs2, 1vs3, 1vs4, 1vs5)

As this table was quite extensive, we decided to provide all comparisons in the supplementary material (table B1) and in the paper only the significant results. We stated this in line 376-377.

Discussion: The authors could draw on other recent articles to discuss their results

-  Chumakov, E., Petrova, N., Mamatkhodjaeva, T., Ventriglio, A., & Bhugra, D. (2022). The impact of covid-19: Anxiety, depression, and wellbeing among medical students. International Journal of Social Psychiatry, 68(6), 1270-1276.

-   Avila-Carrasco, L., Díaz-Avila, D. L., Reyes-López, A., Monarrez-Espino, J., Garza-Veloz, I., Velasco-Elizondo, P., ... & Martinez-Fierro, M. L. (2023). Anxiety, depression, and academic stress among medical students during the COVID-19 pandemic. Frontiers in Psychology, 13, 1066673.

-    AlJhani, S., Alateeq, D., Alwabili, A., & Alamro, A. (2022). Mental health and online learning among medical students during the COVID-19 pandemic: a Saudi national study. The Journal of Mental Health Training, Education and Practice, 17(4), 323-334.

After reading the publications, we implemented the references in the discussion part underlining our assumption of how important the period of data collection is (lines 490-492).

Reviewer 2 Report

Comments and Suggestions for Authors

1.1.  The definition of pre-pandemic was from “2017-2020” in the current study.  As WHO declared the outbreak of COVID-19 on 30 Jan 2020, did you have the cut-off date of 30 Jan 2020 as your pre-pandemic date?  Or would you consider the first lock-down date in Austria as the cut-off please?

1.2.  Similarly, WHO declared the end of COVID-19 pandemic on 5 May 2023, your current peri-pandemic period was from 2022, it is unclear when in 2022.  Please specify.

1.3.  Lines 171-186, you have presented data in the methods section.  I would suggest describing and present data in the results section.

1.4.  Line 194, it is unclear why internal consistency for this sample could not be calculated.  In addition, the internal consistency of other reference data has wide range (0.57 to 0.94).  How did you judge and compare your data with the reference data?

1.5.  It may be good to provide the different assessment tools used in the supplementary files/appendix.  Currently, only COVID-questions were listed in Appendix.

1.6.  Did you assess whether students received any support (both emotional and non-emotional) from the university, other organizations, non-government organizations and social support groups?

1.7.  Do the students have similar socioeconomical status?  Received financial supports vs. no financial supports from any sources?  Did they lose any financial support during COVID such as part-time jobs or parents’ incomes etc?

1.8.  Over 45% of your respondents were first semester students, do you think it would impact on your study as students were starting a new chapter in medical school?

1.9.  In addition, less senior students responded to your survey, what are the potential reasons?  Did they have their own supporting system during the challenging time, or they would rather not to discuss?

1.10.               What kind of drugs are your referring in your “other drugs in comparison”?  Please specify.  Do you include both social drugs, prescriptive and non-prescriptive drugs?

1.11.               Did medical school provide any additional support to students during COVID-19 pandemics?  If yes, please specify.

Comments on the Quality of English Language

It may be good to consider using professional editorial service to improve the current manuscript.

Author Response

Comments and Suggestions for Authors

1.1.  The definition of pre-pandemic was from “2017-2020” in the current study.  As WHO declared the outbreak of COVID-19 on 30 Jan 2020, did you have the cut-off date of 30 Jan 2020 as your pre-pandemic date? Or would you consider the first lock-down date in Austria as the cut-off please?

We considered the first lock-down in Austria as the cut-off date, beginning on March 16th 2020. Data collection started on January 16th and ended on February 20th 2020. We added this information to the text; please see lines 37/38, 174 and 177-178.

1.2.  Similarly, WHO declared the end of COVID-19 pandemic on 5 May 2023, your current peri-pandemic period was from 2022, it is unclear when in 2022.  Please specify.

We added the respective information that data were collected in March 2022 (line 178).

1.3.  Lines 171-186, you have presented data in the methods section.  I would suggest describing and present data in the results section.

Thank you for this suggestion, we moved or rather divided the respective parts to the results section – they can now be found within the cross-sectional or longitudinal analyses as well as in the statistical analyses chapter.

1.4.  Line 194, it is unclear why internal consistency for this sample could not be calculated.  In addition, the internal consistency of other reference data has wide range (0.57 to 0.94).  How did you judge and compare your data with the reference data?

As we already pointed out in the manuscript: “The standardized Mental Health Composite Score (MCS) results from all 12 items (raw data ranging from 0 to 100) being differently weighted by various regression coefficients and summed up” (lines 188-190).

Therefore, the calculation of the SF-12 reliability is very difficult due to the specific characteristics of the scale since it requires reliability estimates for each of the underlying eight subscales (originally emanating from the SF-36), some of which has only one item. Moreover, the different weighting of each individual item in relation to the total scale and the different response formats of the subscales are problematic (Wirtz, Morfeld & Glaesmer, 2017). It is assumed that reliability can be most likely calculated by computing weighted composite scores (Ruotolo et al., 2021), but this procedure is complicated as the above-mentioned reliability estimates of all subscales would be needed (Ware & Kosinsiki, 2003; Ruotolo et al., 2021). Summing up, Cronbach's alpha values reported in the literature represent only an estimate of true reliability (Ruotolo et al., 2021). Under these circumstances, we decided to refer to existing reference data.

We followed your suggestions and (1) tried to clarify why Cronbach’s alpha is so difficult to calculate and (2) updated the MCS reference data according to the results of Ruotolo et al. (2021), who investigated medical and health professionals' students (lines 191-195). They reported a Cronbach's alpha of .84 for the MCS and with this reference we assume sufficient reliability of the SF-12 scale for our sample as well.

Ruotolo, I., Berardi, A., Sellitto, G., Panuccio, F., Polimeni, A., Valente, D., & Galeoto, G. (2021). Criterion Validity and Reliability of SF-12 Health Survey Version 2 (SF-12v2) in a Student Population during COVID-19 Pandemic: A Cross-Sectional Study. Depression research and treatment, 2021, 6624378. https://doi.org/10.1155/2021/6624378

Ware, J.E. & Kosinski, M. (2003). SF-36 Physical and Mental Health Summary Scales: A Manual for Users of Version 1. Second Edition, Lincoln, RI: QualityMetric Incoroporated.

Wirtz, M., Morfeld, M., Glaesmer, H. & Brähler, E. (2017). Konfirmatorische Prüfung der Skalenstruktur des SF-12 Version 2.0 in einer deutschen bevölkerungsrepräsentativen Stichprobe. Diagnostica. 64. 1-13. 10.1026/0012-1924/a000194.

1.5.  It may be good to provide the different assessment tools used in the supplementary files/appendix.  Currently, only COVID-questions were listed in Appendix.

Besides the SF-12 (licensed tool), all tools are either open access or can be received on request by the authors (please follow the respective references). All applied questionnaires / assessment tools are described in the 2.2. Measures section including the same construct abbreviations used in the Results section in the tables and in the supplementary file. Moreover, due to copyright issues, we do not reprint any original material.

1.6.  Did you assess whether students received any support (both emotional and non-emotional) from the university, other organizations, non-government organizations and social support groups?

We assessed ‘social support’ from the German ‘Salutogenetic Subjective Working Analysis’ (SALSA) comprising if there is (1) someone you can rely on if study problems appear, (2) willingness of someone to listen to your problems, and (3) someone who actively supports you to facilitate your study, each related to lecturers, study colleagues and persons outside university (e.g., partner, friends, family). We also report on the respective results, please see table 2/3 or lines 336, 344, 401, etc.

1.7.  Do the students have similar socioeconomical status?  Received financial supports vs. no financial supports from any sources?  Did they lose any financial support during COVID such as part-time jobs or parents’ incomes etc?

There was no specific question on the students’ socioeconomic status or if they received/lost any financial support, however, we asked in general if they have a job beside their study (and how many hours they spend on it) and if they suffer from any financial problems due to the COVID environment. We added the respective information; please see table 1 and lines 305-307 as well as 394-396.

1.8.  Over 45% of your respondents were first semester students, do you think it would impact on your study as students were starting a new chapter in medical school?

We already stated that first year students felt better informed about the prevention measures at university than later year students and that beginners have not experienced former study arrangements (lines 476-481) also entailing their ‘better’ results. However, in particular 2nd year students pre-pandemic showed the most differences compared to 2nd year students peri-pandemic, with the latter reporting fewer burnout symptoms and stressors, more social support of lecturers and study satisfaction (lines 343-346) also listed in table 3 (and the supplementary file) reflecting possible effects of the respective study years pre- and peri-pandemic.

1.9.  In addition, less senior students responded to your survey, what are the potential reasons?  Did they have their own supporting system during the challenging time, or they would rather not to discuss?

As we survey all medical students annually, senior students ‘know’ our study already and may be less interested in answering +/- ‘the same questions’ again. Furthermore, the test battery is quite extensive and we assume that they are more involved in practical courses and clinical internships. We experience this phenomenon over the whole survey with lower responding rates in higher semesters. However, in the longitudinal design more senior students responded (please see table 1).

1.10.    What kind of drugs are you referring in your “other drugs in comparison”?  Please specify.  Do you include both social drugs, prescriptive and non-prescriptive drugs?

The question refers to the use of cannabis etc. (please see question III in the appendix). In the German language, there cannot be any misunderstanding regarding the term “drugs”.

1.11. Did medical school provide any additional support to students during COVID-19 pandemics?  If yes, please specify.

Besides the usual student support for mental health, no COVID-specific support has been offered by the Medical University.

Comments on the Quality of English Language

It may be good to consider using professional editorial service to improve the current manuscript.

We ordered an English proof reading using the professional service linked on the website of the “International Journal of Environmental Research and Public Health” (MDPI).

Reviewer 3 Report

Comments and Suggestions for Authors

I would like to thank All Authors and editors for the opportunity to revise the paper titled “Well-being, Mental Health and Study Characteristics of Medical Students Before and During the Pandemic”.

The study is about the well-being and burnout of medical students evaluated pre and post-pandemic.

The study is very interesting, but I would like the Authors to clarify some aspects. Therefore, I think that the manuscript could be considered the publication after a minor revision.

Introduction

I think that the Authors could have considered more recent literature about the mental health of medical students and with respect to European Countries during the pre-pandemic period. , I suggest:

Leombruni P, et Al. Stress in Medical Students: PRIMES, an Italian, Multicenter Cross-Sectional Study. Int J Environ Res Public Health. 2022 Apr 20;19(9):5010. doi: 10.3390/ijerph19095010. PMID: 35564409; PMCID: PMC9100187.

Bert F, et Al. Prevalence of depressive symptoms among Italian medical students: The multicentre cross-sectional "PRIMES" study. PLoS One. 2020 Apr 17;15(4):e0231845. doi: 10.1371/journal.pone.0231845. PMID: 32302354; PMCID: PMC7164645.

Materials and Methods

I suggest reporting only the cross-sectional data, first because it's very difficult to read an article reporting results from two different study designs, secondary the longitudinal data as the authors reported are strongly limited.

I suggest simplifying the manuscript or eventually reporting as a secondary analysis the results of longitudinal data. In this way, the reader could read the results on the tables and in the text easily. 

I think that the information about the semester could be not reported. 

Discussion 

With respect to satisfaction, I suggest an Italian study that includes medical students: Cofini V, et Al. E-Learning Satisfaction, Stress, Quality of Life, and Coping: A Cross-Sectional Study in Italian University Students a Year after the COVID-19 Pandemic Began. Int J Environ Res Public Health. 2022 Jul 5;19(13):8214. doi: 10.3390/ijerph19138214. PMID: 35805872; PMCID: PMC9266753.

Author Response

Comments and Suggestions for Authors

I would like to thank All Authors and editors for the opportunity to revise the paper titled “Well-being, Mental Health and Study Characteristics of Medical Students Before and During the Pandemic”. The study is about the well-being and burnout of medical students evaluated pre and post-pandemic. The study is very interesting, but I would like the Authors to clarify some aspects. Therefore, I think that the manuscript could be considered the publication after a minor revision.

Thank you very much for your valuable review. Please find our answers and revisions made in the paper as follows.

Introduction

I think that the Authors could have considered more recent literature about the mental health of medical students and with respect to European Countries during the pre-pandemic period. I suggest:

Leombruni P, et Al. Stress in Medical Students: PRIMES, an Italian, Multicenter Cross-Sectional Study. Int J Environ Res Public Health. 2022 Apr 20;19(9):5010. doi: 10.3390/ijerph19095010. PMID: 35564409; PMCID: PMC9100187.

Bert F, et Al. Prevalence of depressive symptoms among Italian medical students: The multicentre cross-sectional "PRIMES" study. PLoS One. 2020 Apr 17;15(4):e0231845. doi: 10.1371/journal.pone.0231845. PMID: 32302354; PMCID: PMC7164645.

Thank you for highlighting these publications. Within the introduction, we replaced one U.S. related study with Lemobruni et al. [6]. In terms of the chapter “Mental health”, we decided to stay with the global prevalence rates of anxiety and depression before and during the pandemic (as the COVID-pandemic has been a global issue) presented by the cited meta-analyses including various European countries. Your recommended articles refer exclusively to Italian medical students and do not include other European countries as you suggest emphasizing in our paper. Nevertheless, we refined the respective part by providing the prevalence rates for Europe (lines 95-98).

Materials and Methods

I suggest reporting only the cross-sectional data, first because it's very difficult to read an article reporting results from two different study designs, secondary the longitudinal data as the authors reported are strongly limited. I suggest simplifying the manuscript or eventually reporting as a secondary analysis the results of longitudinal data. In this way, the reader could read the results on the tables and in the text easily. I think that the information about the semester could be not reported. 

We agree that combining cross-sectional and longitudinal data in one publication might be more challenging and we are aware of the limitations. However, we would like to keep these longitudinal data in the paper and share them for the first (and only) time, as that data set will not grow any further in terms of COVID-specific information. Formally, some data from 2023 might add a little but when looking at the medical universities’ schedules, COVID was no longer noticeable in 2023. Moreover, the WHO declared the end of the pandemic on 5th May 2023 and we think that this is the best opportunity to publish our matched data before the topic runs out. The information on the semesters will also remain in the paper as other reviewers appreciated them. However, we tried to clarify the structure of the paper to improve readability (e.g., separating sample descriptions for cross-sectional and longitudinal analyses; lines 303-307 and 389-396).

Discussion 

With respect to satisfaction, I suggest an Italian study that includes medical students: Cofini V, et Al. E-Learning Satisfaction, Stress, Quality of Life, and Coping: A Cross-Sectional Study in Italian University Students a Year after the COVID-19 Pandemic Began. Int J Environ Res Public Health. 2022 Jul 5;19(13):8214. doi: 10.3390/ijerph19138214. PMID: 35805872; PMCID: PMC9266753.

Thank you for sharing this publication. In our study, we refer to study satisfaction in general (e.g., supervision of lecturers, the professional quality of the courses, the study structure, number of participants in the courses, etc. – please see lines 440-442) possibly also including e-learning to a certain extent, however, we do not explicitly go into the topic of “e-learning”. Therefore, we think that discussing both constructs would be too extensive but we added the idea of the paper (lines 424-425).

Reviewer 4 Report

Comments and Suggestions for Authors

The work entitled “Well-being, mental health and study characteristics of medical students before and during the pandemic” is interesting. The effects of the pandemic on human health, not only physical but also mental, will undoubtedly be observed for a very long time.

The work requires editing to create a concise, logical whole. It is essential to characterize the research group and justify its selection correctly. With a relatively large number of different statistical analyses, it is also important to clearly explain the statistical analysis. For example, it is not enough to indicate in lines 169-170: "To calculate the difference between multiple groups in interval-scale". Please explain "multiple groups" in detail.

Selected detailed comments.

Please read the work carefully and complete the missing literature references (e.g. lines:84-87). I also propose removing fragments that duplicate information presented by the authors.

There are fragments in which it needs to be clarified whether the authors are describing the condition of students before or during Covid-19. This can only be guessed at. The same applies to students in general, including medical students.

I propose to refer in the introduction to the relationship between Well-being, mental health, and sociodemographic data that the authors used in the work (Table 1), including living situation and family status.

Line 162 Please explain why one of the two criteria of the people taking part in the study was "fluently speaking German"? How could this be assessed based on an online survey?

Lines 171-178. Please redact the indicated fragment so there are no doubts about how the research sample was selected. As it stands, it is unclear and questionable.

Please also consider rewording the "Limitations" part. Do the authors of the work themselves question the point of conducting them?

Author Response

Comments and Suggestions for Authors

The work entitled “Well-being, mental health and study characteristics of medical students before and during the pandemic” is interesting. The effects of the pandemic on human health, not only physical but also mental, will undoubtedly be observed for a very long time. The work requires editing to create a concise, logical whole. It is essential to characterize the research group and justify its selection correctly.

With a relatively large number of different statistical analyses, it is also important to clearly explain the statistical analysis. For example, it is not enough to indicate in lines 169-170: "To calculate the difference between multiple groups in interval-scale". Please explain "multiple groups" in detail.

Thank you for your suggestion. We tried to clarify the assignment of the research groups (please see lines 285-288) and refined the statistical analyses as well as the description of the multiple groups (lines 269-279).

Selected detailed comments.

Please read the work carefully and complete the missing literature references (e.g. lines: 84-87). I also propose removing fragments that duplicate information presented by the authors.

We added the missing literature reference by re-citing the website of the WHO for the mental health definition [23]. Furthermore, we re-cited [3] etc. to complete possible missing links and added references on the medical study conditions [11,12], the governmental vaccination mandate [17] and sociodemographic data linked to well-being and mental health [28], which was suggested as well. Other references (marked in red) were added in terms of other discussion parts/reviews. Furthermore, we deleted ‘doubled’ information about general population and COVID-19, burnout prevalence rates, and the JD-R model in chapter 1 (introduction), data collection and assignment of students to the pre- and peri-pandemic group in chapter 2.3 (statistical analyses), and pre- vs- peri-pandemic phrases in chapter 3 (results).

There are fragments in which it needs to be clarified whether the authors are describing the condition of students before or during Covid-19. This can only be guessed at. The same applies to students in general, including medical students.

We carefully went through the whole paper and tried to clarify the relations (lines 98, 107, 122, 123, 125, 142, 143).

I propose to refer in the introduction to the relationship between Well-being, mental health, and sociodemographic data that the authors used in the work (Table 1), including living situation and family status.

We added a short information on that in the introduction (line 104-106), however, as we do not statistically investigate these sociodemographic data in terms of our measured constructs and just provide sample characteristics, we will not deepen that issue and stick to the contents included into the analyses.

Line 162: Please explain why one of the two criteria of the people taking part in the study was "fluently speaking German"? How could this be assessed based on an online survey?

Thank you for pointing this out. The official language of the Medical University is German with all mandatory lectures and courses being held in German. Therefore, both the first-time admission (‘MedAT’ test) and the continuation of medical studies require German language skills at level C1 (please see line 46-47). Therefore, we deleted this “criteria”.

Lines 171-178. Please redact the indicated fragment so there are no doubts about how the research sample was selected. As it stands, it is unclear and questionable.

We completely re-worked this paragraph, and shifted parts to the statistical analyses, the cross-sectional and longitudinal analyses and deleted misleading information.

Please also consider rewording the "Limitations" part. Do the authors of the work themselves question the point of conducting them?

We are definitely not questioning our work per se, but we did not want to underestimate certain issues that came up when analysing all the different facets. However, we tried to re-phrase this part accordingly.

Reviewer 5 Report

Comments and Suggestions for Authors

Dear Authors,

The manuscript entitled “Well-being, mental health and study characteristics of medical students before and during the pandemic” deals with an interesting and important topic, the manuscript is of high-quality. I have only a few minor recommendations to deal with.

In the abstract you mention “sub-study” but it is not clear why it is a “sub”-study (what is the main study?). It would be advantageous if you could include data on the well-being of the general population in the Introduction section. There is an excess bracket in line 167. Why did ‘Patient Health Questionnaire’ replace the SF-12 in 2022? Please, explain the DSM-IV (line 201). Why did you classify year 2020 as ‘pre-pandemic’, when the pandemic started at the beginning of 2020 and the strictest measures were implemented in 2020? Please revise the sentences in lines 337-340. In line 351, it would be better to use “less or at most equal amounts.”

Author Response

Dear Authors,

The manuscript entitled “Well-being, mental health and study characteristics of medical students before and during the pandemic” deals with an interesting and important topic, the manuscript is of high-quality. I have only a few minor recommendations to deal with.

In the abstract you mention “sub-study” but it is not clear why it is a “sub”-study (what is the main study?).

Thank you for highlighting this, we called it a ‘sub-study’ as not all available data found their way into this analysis due to various reasons: e.g., data from 2015 and 2016 were not taken into account as none of these students participated in 2022, or they were already published many times, and the officially funded project time was already over. We agree on a potential confusion by that, deleted that phrase or rather all information that could be misleading, and changed it into ‘study’.

It would be advantageous if you could include data on the well-being of the general population in the Introduction section.

We already cited two studies in the introduction referring to the general population [4,5] in terms of psychological well-being and negative mental health outcomes during the pandemic (lines 38-41). We think that going more into detail with data of the general population would shift the focus of this paper somehow and as we are not comparing medical students with the general population, we think this information is sufficient.

There is an excess bracket in line 167.

We deleted this bracket.

Why did ‘Patient Health Questionnaire’ replace the SF-12 in 2022?

Due to many reviews concerning former publications, we decided that the PHQ and the GAD are more suitable to assess mental health (or possible clinical cases of depression/anxiety). In particular, for students, a rather young and vital group, physical health or limitations measured with the SF-12 in terms of, e.g., not being able to climb some stairs, or having massive pain influencing them, was mostly not significant. Moreover (as we describe in our paper), most studies reporting on mental health use specific measurements like the PHQ or the GAD, so comparison with our data is now possible.

Please, explain the DSM-IV (line 201).

We changed it to ‘Diagnostic and Statistical Manual of Mental Disorders, 4th edition’ (DSM-IV); line 200-201.

Why did you classify year 2020 as ‘pre-pandemic’, when the pandemic started at the beginning of 2020 and the strictest measures were implemented in 2020?

Our data from 2020 were assessed from January 2020 to the beginning of March 2020 before the first lockdown in Austria (March 16th, lines 37-38). Therefore, we subsumed these data under the term ‘pre-pandemic’. To avoid any confusion we added the period of data collection (lines 177-178).

Please revise the sentences in lines 337-340.

We revised that part.

In line 351, it would be better to use “less or at most equal amounts.”

We followed your suggestion.

Round 2

Reviewer 4 Report

Comments and Suggestions for Authors

The article has been significantly improved. According to the reviewer, the statistical analysis of the results is still questionable.

Author Response

Dear reviewer,

the academic editor provided your feedback and the STROBE checklist.
We have re-worked the methods section as concise as possible following this checklist. However, in terms of readability, we combined some points of the checklist.
As we invited all active medical students to take part we neither see any selection bias a-priori nor did we use any specific sampling method. We refer to possible biases a-posteriori based on the response rate or the "missing" year of data collection 2021 in the limitation section (please see lines 644-650).

Finally, the MDPI English language editing as recommend by another reviewer is done and we will upload the new file.

Thank you.